# Region-Specific Effects of Fractionated Low-Dose Versus Single-Dose Radiation on Hippocampal Neurogenesis and Neuroinflammation

**DOI:** 10.3390/cancers14225477

**Published:** 2022-11-08

**Authors:** Zoé Schmal, Claudia E. Rübe

**Affiliations:** Department of Radiation Oncology, Saarland University Medical Center, 66421 Homburg, Germany

**Keywords:** radiotherapy, radiosensitivity, hippocampus, neurogenesis, neuroinflammation

## Abstract

**Simple Summary:**

Radiotherapy-associated brain injury with neurocognitive impairment is a common long-term side effect for brain cancer survivors, affecting the quality of life particularly in pediatric patients. The complex pathophysiology of hippocampus-dependent cognitive deterioration with respect to fractionated radiotherapy and the precise role of neurogenesis in radiation-induced neuroinflammation is poorly understood. In a mouse model with fractionated low-dose radiation, we analyzed different hippocampal subregions to precisely elucidate the underlying mechanisms of radiation-induced brain injury. Our findings suggest that region-specific differences in radiosensitivity are mainly based on the presence of proliferating neuroprogenitors. Even low single doses to the neurogenic niche of the hippocampus lead to neuronal damage with subsequent neuroinflammation. Thus, limiting both cumulative doses and dose fractions to the hippocampal stem cell niche is an important issue of clinical radiotherapy to preserve neurocognitive functions.

**Abstract:**

Background: Despite technical advances in hippocampus-sparing radiotherapy, radiation-induced injury to neural stem cell compartments may affect neurocognitive functions. In pre-clinical mouse models with fractionated low-dose radiation (FLDR) and single-dose radiation (SDR), the accurate response to radiation-induced injury was analyzed in different hippocampal subregions. Methods: Adult and juvenile C57BL/6NCrl mice were exposed to FLDR (20 × 0.1 Gy, daily exposure from Monday to Friday for 4 weeks) or SDR (1 × 2 Gy). In addition, 72 h after the last exposure, neuroglia (astrocytes and microglia) and neuroprogenitor cells were characterized and quantified in the hippocampal cornu ammonis (CA) and dentate gyrus (DG) by immunofluorescence studies. Results: After analyzing different hippocampal subregions, it was observed that radiation responses varied between non-neurogenic CA, with no detectable inflammatory alterations, and neurogenic DG, characterized by impaired neurogenesis and subsequent neuroinflammation. Age-dependent differences in radiosensitivity appeared to depend on the varying proliferative potential of neural stem cell niches. Using the same overall dose for FLDR and SDR (2 Gy), both the cumulative dose over time and also the single dose fraction have decisive impacts on hippocampal damage. Conclusion: Region-specific effects of radiation-induced hippocampal injury relies primarily on cell deaths of proliferating neuroprogenitors. Dose per fraction defines the extent of neuronal injury, and subsequently activated microglia and reactive astrocytes modulate dynamic processes of neuroinflammation. Thus, limiting both cumulative doses and dose fractions to hippocampal DG is an important issue of clinical radiotherapy to preserve neurocognitive functions.

## 1. Introduction

High-precision radiotherapy of brain tumors facilitates the delivery of therapeutic doses to target volumes, while reducing the exposure of surrounding healthy brain tissue to low doses. Despite major technical advancements in medical imaging and radiotherapy planning, cancer survivors often suffer disabling cognitive dysfunction [1]. These neurotoxic effects of ionizing radiation (IR) are particularly harmful for the developing brain of children and adolescents [2]. Abundant evidence suggests that the hippocampus is one of the most radiosensitive regions of the brain. To minimize radiation-induced damage of healthy tissues, radiotherapy is usually “fractionated”, which means that the total radiation dose is divided into multiple “fractions” over the treatment course, lasting multiple weeks. However, the effects of fractionation on the highly vulnerable hippocampus region have not yet been analyzed in detail.

Hippocampal formation is crucial for memory processing, learning, spatial navigation, and emotions [3]. The hippocampus is divided into the dentate gyrus (DG) and different subregions of the cornu ammonis (CA1-CA4). In the subgranular zone (SGZ) of the DG, neural stem cells continuously self-renew and differentiate into neurons in a process called *adult neurogenesis* [4]. Depending on the chronological age of the individual, new neurons are generated from asymmetrical division of progenitor radial glial cells in the SGZ, a narrow layer of cells located between the granule cell layer (GCL) and the *hilus* of the DG. During their post-mitotic maturation, these neuroprogenitors of the SGZ migrate into adjacent GCL, where they establish their mature morphological and functional characteristics with the outgrowth of axons and dendrites, thereby integrating themselves into established neuronal networks [5]. Increasing amounts of evidence indicate that adult neurogenesis is tightly controlled by environmental conditions in the neurogenic niche, which consists of glia cells, such as microglia and astrocytes. Microglia cells are the immune cells of the CNS, and consequently play important roles in brain infections and inflammation. Microglia are specialized for the uptake and removal of pathogens, apoptotic cells and cellular debris [6]. Astrocytes are the most numerous cell type within the brain and perform a variety of tasks, from axon guidance and synaptic support, energy delivery to neurons, to the control of the blood–brain barrier [7]. Collectively, glial cells maintain the homeostatic environment of the neurogenic niche to support the development of adult-born neurons in healthy brains [8]. However, under pathological conditions such as radiation-induced injury [9], microglia and astrocytes play a pivotal role in the modulation of neuroinflammation [10].

Our recent experimental studies suggest that even low doses of IR contribute to cognitive decline by disturbing hippocampal neurogenesis and inducing chronic neuroinflammation [11,12]. Here, in a pre-clinical model with adult and juvenile mice, we analyzed radiation-induced changes in different hippocampal subregions, following fractionated low-dose versus single-dose irradiation, to evaluate the effect of fractionation on adult neurogenesis and brain homeostasis.

## 2. Materials and Methods

### 2.1. Animal Radiation and Tissue Sampling

Male C57BL/6NCrl mice were obtained from Charles River Laboratories (Sulzfeld, Germany) and were housed in groups of 3 to 6 animals in IVC cages, under standard laboratory conditions (22 °C ± 2, 55% ± 10 humidity, 12 h:12 h light/dark cycle; ad libitum feeding conditions). For whole-body radiation with a linear accelerator (Artiste™, Siemens Healthineers, München, Germany), mice were placed in Plexiglas cylinders covered with 1.5 cm thick tissue-equivalent material to improve dose homogeneity. The following conditions were used: radiation field: 0.3 m × 0.3 m; collimator and gantry angle: 0°: source-surface distance: 2 m for 0.1 Gy, 1 m for 2 Gy; beam energy: 6 MV photons; dose rate: 2 Gy/min. Tomography-based, 3D-dose calculations were performed using the Pinnacle™ planning system (Philips Radiation Oncology Systems, Fitchburg, WI, USA). Thermoluminescent dosimetry was used to confirm the reliable and uniform delivery of 0.1 Gy and 2 Gy, respectively. For FLDR experiments, juvenile and adult C57BL/6NCrl mice (age of P11 or P56 at start of FLDR) were daily irradiated from Monday to Friday for 4 weeks. For SDR experiments, adult C57BL/6NCrl mice (age of P56) were irradiated once with 2 Gy. In addition, 72 h after (the last) exposure, animals were anesthetized, transcardially perfused with 4% paraformaldehyde, and their brains were removed and post-fixed in 4% paraformaldehyde. For each examination method, right and left hippocampi of 3 different adult and juvenile mice (n = 3), each with 3 technical replicates (3 tissue sections with an 80 µm distance) were analyzed at each assessment time-point and compared to sham-irradiated age-matched controls. Experimental studies were approved by the Animal Care and Use Committee of the Saarland University.

### 2.2. Immunofluorescence Microscopy (IFM)

Formalin-fixed tissues were embedded in paraffin and cut into 4 μm thick serial sections −1.9 mm from the bregma. After removing paraffin from xylene, sections were rehydrated in decreasing alcohol concentrations. For antigen retrieval, tissues were boiled in citrate buffer and pre-incubated with Roti™-Immunoblock (Carl Roth, Karlsruhe, Germany) or 2% goat serum. Subsequently, coronal sections obtained 80µm apart (every 20th section) were incubated with primary antibodies (GFAP, SA100β, IBA-1, DCX: Abcam, Cambridge, UK), followed by incubation with Alexa-Fluor-488 or Alexa-Fluor-568 secondary antibodies (Invitrogen, Karlsruhe, Germany). Finally, sections were mounted in VECTAshield™ with 4′, 6-diamidino-2-phenylindole (DAPI; Vector Laboratories, Burlingame, CA, USA). A Nikon Eclipse Ni-E epifluorescent microscope equipped with a charge-coupled device camera (DS-Qi2) and acquisition software (NIS-Elements™, Nikon, Düsseldorf, Germany) was used for sample analysis and image capturing. Positive cells were counted in the subgranular zone (SGZ), within the dentate hilus and granule cell layer (GCL), whose area was measured using NIS-Elements™ software (version 4.300; Nikon, Düsseldorf, Germany). A minimum of three different mice with three technical replicates were screened for GFAP+/S100β+ (astrocytes), IBA1+ (microglia) and DCX+ cells (neuroprogenitor). Quantifications were performed bilaterally in both hippocampi and cell counts per area (cell/mm^2^) were determined.

### 2.3. Statistical Analysis

All statistical analyses were performed using the statistical software Graphpad Prism. Normally distributed data were presented as mean ± SD and differences between groups were determined by unpaired T-Tests. Statistical significance was presented as * *p* < 0.05, ** *p* < 0.01, *** *p* < 0.001.

## 3. Results

### 3.1. Microglia and Astrocytes in the Cornu Ammonis (CA) after Fractionated Low-Dose and Single-Dose Radiation

Activation of glial cells plays an important role in the pathogenesis of radiation-induced brain injury [6]. Within healthy brains, ramified microglia are highly active, continuously extending and retracting their fine processes, thereby scanning their microenvironment for pathogens. Astrocytes are the most numerous and diverse neuroglial cells in the CNS and their heterogeneous mmorphologies depend on their actual activation status. To gain an overview of the potential inflammatory reactions following fractionated low-dose radiation (FLDR) versus single-dose radiation (SDR), we started to examine microglia cells and astrocytes in the hippocampal CA. Representative IF micrographs show the distribution of IBA1-labeled microglia and GFAP-/S100β-labeled astrocytes after FLDR (20 × 0.1 Gy, 72 h post-IR) and SDR (1 × 2 Gy, 72 h post-IR) compared to sham-irradiated controls (Figure 1). Following FLDR and SDR, we observed some activated microglia in the non-neurogenic CA region, but no clear differences in the number of glia cells compared to sham-irradiated, age-matched controls.

### 3.2. Microglia in the Dentate Gyrus (DG) after Fractionated Low-Dose and Single-Dose Radiation

Under physiological conditions, resting microglia populations are characterized by elaborated thin processes with multiple branches (Figure 2A, upper row). Following brain insults, the morphology of activated microglia changes dynamically, depending on the type and severity of the stimuli. Activated microglia retract their processes, which become fewer and thicker during their stress response and disappear in the fully activated state [13]. In response to radiation-induced injury, the density of microglia cells with activated phenotypes clearly increased at 72 h post-IR (Figure 2A). Following FLDR and SDR, microglia numbers were also significantly elevated at 72 h post-IR for FLDR (juvenile and adult hippocampi) and SDR compared to sham-irradiated, age-matched controls (Figure 2B). While microglia levels after FLDR were elevated by ≈40% at 72 h post-IR (Figure 2B, juvenile (P39): sham-IR: 120 ± 14 cells/mm^2^; 72 h post-IR: 167 ± 13 cells/mm^2^; adult (P84): sham-IR: 110 ± 9 cells/mm^2^, 72 h post-IR: 156 ± 10 cells/mm^2^), even greater increases of nearly ≈50% at 72 h post-IR were observed after SDR (Figure 3B, sham-IR: 154 ± 13 cells/mm^2^, 72 h post-IR: 228 ± 9 cells/mm^2^) (Figure 2B). Collectively, this increase in microglial activation suggests radiation-induced inflammatory responses.

### 3.3. Astrocytes in the Dentate Gyrus (DG) after Fractionated Low-Dose and Single-Dose Radiation

Astrocytes exhibit morphological and functional alterations in response to traumatic brain injury [10]. Following repetitive insults by FLDR, the reactive astrocytes displayed typical star-shaped morphologies with radially projected, highly branched cellular processes (Figure 3A, second row). During this chronic brain injury, quantification of GFAP+/S100β+ cells revealed significant increases (≈35%) in the number of astrocytes in juvenile (P39: sham-IR: 272 ± 14 cells/mm^2^; 72h post-IR: 368 ± 39 cells/mm^2^) and adult hippocampi (P84: sham-IR: 304 ±24 cells/mm^2^, 72 h post-IR: 411 ± 14 cells/mm^2^) (Figure 3B). Following SDR with a dose of 2 Gy, by contrast, the number of astrocytes did not increase compared to the sham-irradiated controls, suggesting that astrocytes respond to acute radiation injury with delayed proliferation changes (Figure 3B).

### 3.4. Neuroprogenitors in the Dentate Gyrus (DG) after Fractionated Low-Dose and Single-Dose Radiation

In sham-irradiated hippocampi, numerous doublecortin-expressing (DCX+) neuroprogenitors in the SGZ of DG showed intact dendritic arborizations with processes that extended across granular cell layers (Figure 4A, upper row). In Figure 4A, representative micrographs of fractionated low-dose (20 × 0.1 Gy, 72 h post-IR) and single-dose (2 Gy, 72 h post-IR) irradiated hippocampi are depicted to demonstrate the directly visible neurotoxic effects (Figure 4A, lower rows). Following FLDR, the hippocampal DG exhibited neuroprogenitors with dramatically impaired dendritic arborization (Figure 4A second row), indicating the critical impact of even low doses on neuronal differentiation and maturation. Furthermore, the number of DCX+ neuroprogenitors was significantly reduced (juvenile P39: sham-IR: 915 ± 52 cells/mm^2^; 72 h post-IR: 593 ± 23 cells/mm^2^, ≈35% reduction; adult P84: sham-IR: 398 ± 25 cells/mm^2^; 72 h post-IR: 305 ± 24 cells/mm^2^, ≈25% reduction) (Figure 4B). Following SDR, the population of neural progenitors in hippocampal DG was even more drastically reduced, so that nearly all cell bodies of DCX+ neuroprogenitors were completely abrogated (P56: sham-IR: 779 ± 91, 72 h post-IR: 109 ± 29 cells/mm^2^, ≈85% reduction) (Figure 4A,B). Collectively, these findings demonstrate the fatal effects of ionizing radiation on the hippocampal DG, leading to drastically reduced neurogenesis, neuronal differentiation and maturation.

**Figure 3 cancers-14-05477-f003:**
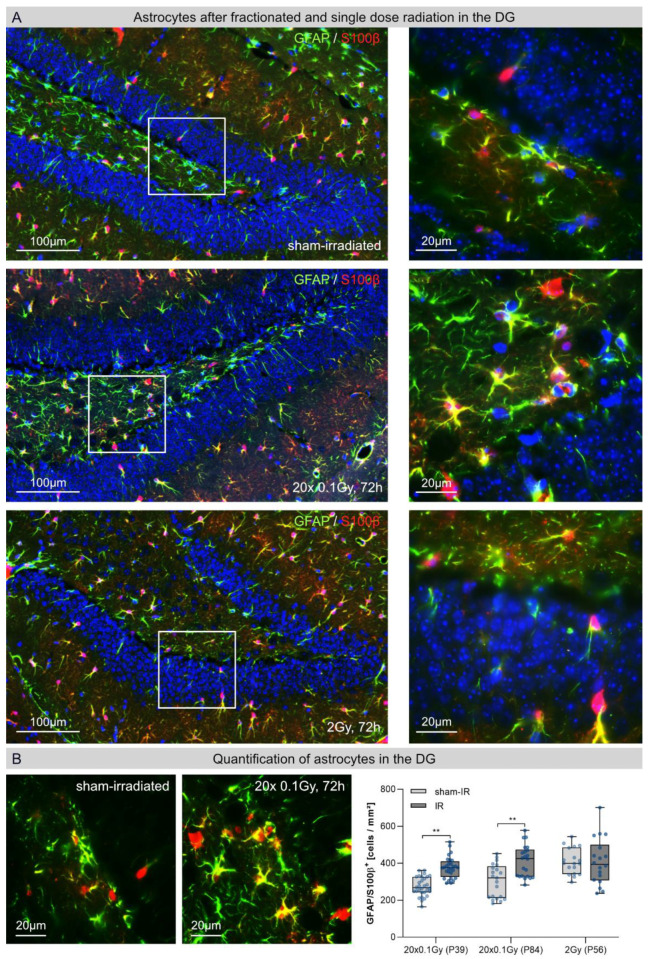
Astrocytes in the hippocampal entate gyrus (DG) after fractionated low-dose and single-dose radiation. (**A**): Immunofluorescence micrographs of GFAP+ (green)/ S100β+ (red) astrocytes in the DG of sham-irradiated (upper panel), fractionated low-dose irradiated (20 × 0.1 Gy, 72 h post-IR; middle panel) and single-dose irradiated (1 × 2 Gy, 72 h post-IR; lower panel) brains. Framed regions are shown at higher magnifications. Insets: following FLDR, reactive astrocytes showed star-shaped morphologies. Original magnification: 600×. (**B**): In the DG of juvenile and adult brains, the numbers of GFAP+/S100β+ astrocytes cells were quantified after 20 × 0.1 Gy versus 1 × 2 Gy (72 h post-IR), in comparison to age-matched, sham-irradiated controls. Error bars represent SEM, n ≥ 3; * denotes statistically significant difference compared to sham-irradiated control: ** *p* < 0.01.

## 4. Discussion

Radiotherapy is the most effective non-surgical treatment of brain tumors and with the improvement in overall survival for these patients over the last few decades, effective strategies that prevent cognitive deterioration after cerebral radiotherapy are required [14]. Radiation-induced injury of the hippocampus is now generally accepted as the main culprit for cognitive deterioration, following cerebral radiotherapy. However, the complex pathophysiology of hippocampus-dependent cognitive dysfunctions and the precise role of neurogenesis in radiation-induced neuroinflammation is still unclear. After analyzing different subregions of the hippocampus, we observed clear differences in the radiation response between the non-neurogenic cornu ammonis (CA), with no detectable radiation-induced inflammatory alterations, and the neurogenic dentate gyrus (DG), characterized by impaired neurogenesis and subsequent neuroinflammation. These results suggest that the hippocampal DG is more vulnerable to radiation-induced damage, presumably as the result of the death of proliferating neuroprogenitors. Moreover, our findings indicate that radiation-induced effects on neurogenesis and neuroinflammation vary between juveniles and adults. This age-dependent difference in radiosensitivity is likely to be attributable to the varying proliferative capacity of the neural stem cell niche. In this preclinical model, our results show that single-dose radiation with 2 Gy (SDR), compared to repetitive low-dose exposures (20 × 0.1 Gy, FLDR), have even more noticeable effects on the development of neuronal network architectures. Dependent on the extent of neuronal injury, local increases in activated microglia and reactive astrocytes modulate the dynamic processes of neuroinflammation and subsequent regeneration. Here, using the same overall dose of 2 Gy, we show that not only cumulative radiation doses over time have a decisive impact on hippocampal damage, but that particularly the dose per fraction defines the biological impact of ionizing radiation on neurogenesis. Collectively, our findings indicate that radiation-induced injury to the hippocampus relies largely on the loss of proliferating neuroprogenitors, with subsequent inflammatory responses of glial cells, to repair the radiation-induced neuronal damage. While SDR with 2 Gy reflects the daily dose-fraction in target volumes, FLDR (20 × 0.1 Gy) may simulate the repetitive lower doses outside the actual target volumes in the context of normo-fractionated conformal radiotherapy. In earlier studies, we have already been able to show that chronic neuroinflammation persists for up to 6 months even following FLDR [11]. Collectively, these results impressively show the well-established fractionation effects and emphasize the importance of lowering the dose fraction for the vulnerable dentate gyrus in the context of clinical radiation planning.

Over the last few years, there has been evidence of growing realization that radiation-induced injury to the hippocampal stem cell compartment inhibits the production and maturation of neuronal progenitors [15], and thus contributes to the development of neurocognitive decline [14]. Previous studies have shown that even very low doses of ionizing radiation results in the progressive decline in hippocampal neurogenesis, with the loss of stem and neuroprogenitor cells and reduced dendritic arborization of hippocampal neurons [11,12]. Higher numbers of stem and progenitor cells in the developing brain enhance this inherent vulnerability to ionizing radiation, and the pronounced neuronal damage explains the enhanced inflammatory reaction in the premature hippocampus [16]. Thus, limiting both the cumulative radiation dose and also the dose fraction to the hippocampal region is an important issue in clinical radiotherapy to preserve neurocognitive functions. In previous work, we were able to show that slowly proliferating SOX2+ stem/progenitor cells survive FLDR and may replace damaged or eliminated DCX+ neuroprogenitor cells through premature differentiation [12]. Therefore, we expect at least partial recovery of hippocampal neurogenesis for both FLDR and SDR with the cumulative dose of 2 Gy. In order to measure functional long-term damage to neurocognitive function, behavioral studies over longer periods of time should be carried out in future experiments.

Clinical oncology prospective studies have analyzed the relationship between RT dose to neural progenitor cell niches and neurocognitive dysfunction in pediatric patients before and 6, 15, and 27 months following completion of cranial RT [15]. These clinical data demonstrated significant associations between increasing RT doses to the hippocampus and decline in neurocognitive skills following cranial irradiation. Accordingly, in recent years, the use of hippocampal sparing techniques to minimize neurocognitive toxicity following cerebral radiotherapy has attracted significant interest [17]. The development of various photon treatment modalities, such as intensity modulated radiation therapy (IMRT), volumetric modulated arc therapy (VMAT) and tomotherapy, offers new options with increased feasibility for hippocampal sparing [18]. Generally, these technical improvements can reduce the hippocampal exposure to a significant extent, but despite optimal radiotherapy planning, the resulting hippocampal doses in clinical settings are considerably higher than the cumulative dose and the dose fraction used in our experimental set-up. Currently, prospective clinical trials are analyzing the feasibility of these techniques in cancer patients with different cranial RT indications and are documenting the oncological and neurocognitive outcomes following hippocampus-sparing radiotherapy [19,20,21,22,23]. However, many studies are still ongoing, so a conclusive assessment is not yet possible.

In conclusion, our findings highlight the extreme radiosensitivity of the hippocampal DG and accentuate the necessity of protecting the hippocampal region by limiting the cumulative dose and by reducing the dose fraction during cerebral radiotherapy to prevent adverse neurotoxic effects.

## Figures and Tables

**Figure 1 cancers-14-05477-f001:**
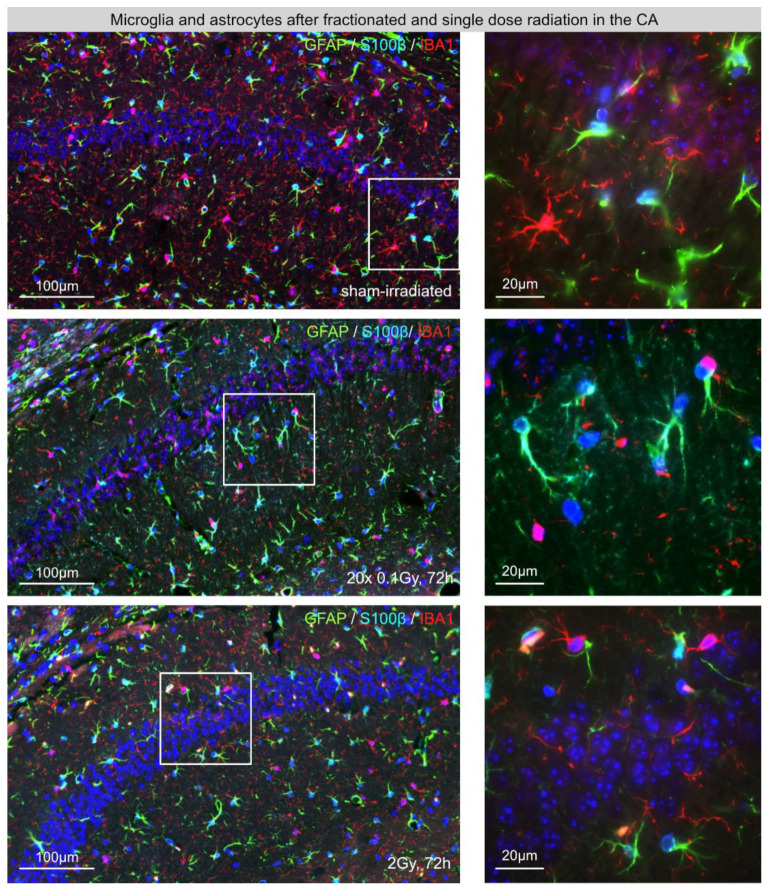
Microglia and astrocytes in the hippocampal cornu ammonis (CA) after fractionated low-dose and single-dose radiation: immunofluorescence micrographs of IBA1+ microglia (red) and GFAP+ (green)/ S100β+ (light-blue) astrocytes in the CA of sham-irradiated (upper panel), fractionated low-dose irradiated (20 × 0.1 Gy, 72 h post-IR; middle panel) and single-dose irradiated (1 × 2 Gy, 72 h post-IR; lower panel) brains. Framed regions are shown at higher magnifications. Insets: following FLDR and SDR, activated microglia retract their processes, which become fewer and thicker during their stress response. Original magnification: 600×.

**Figure 2 cancers-14-05477-f002:**
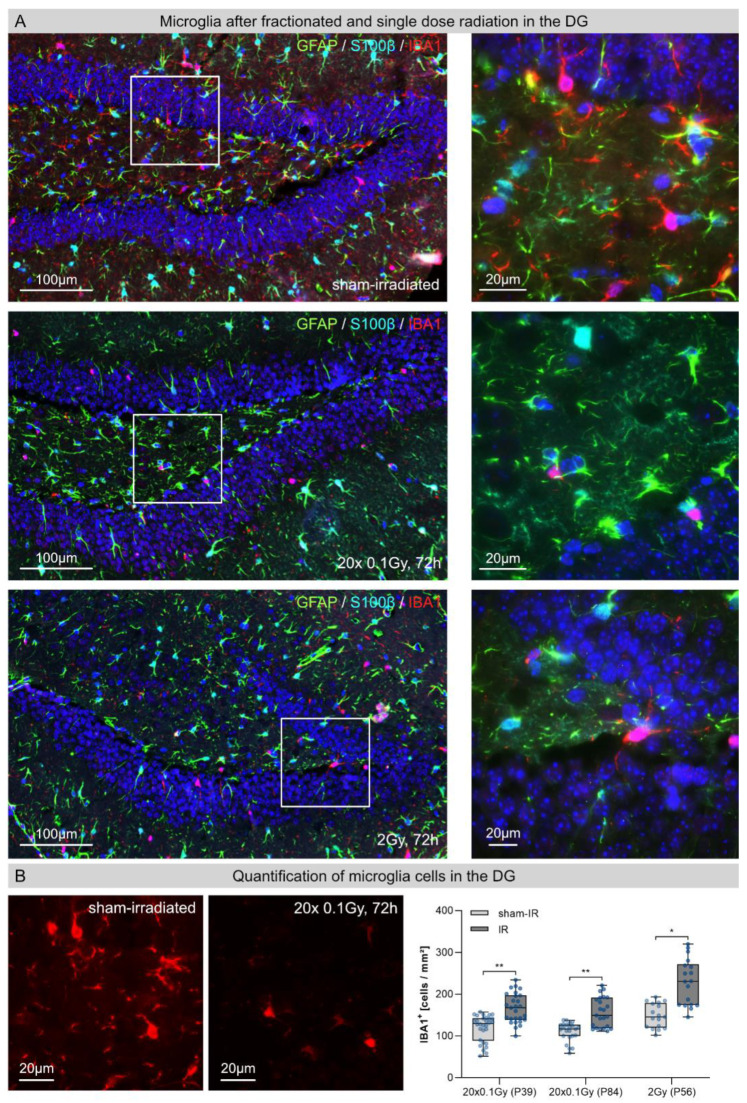
Microglia in the hippocampal dentate gyrus (DG) after fractionated low-dose and single-dose radiation. (**A**): Immunofluorescence micrographs of IBA1+ microglia (red) and GFAP+ (green)/ S100β+ (light blue) astrocytes in the DG of sham-irradiated (upper panel), fractionated low-dose irradiated (20 × 0.1 Gy, 72 h post-IR; middle panel) and single-dose irradiated (1 × 2 Gy, 72 h post-IR; lower panel) brain. Framed regions are shown at higher magnifications. Insets: following FLDR and SDR, activated microglia cells and reactive astrocytes were characterized by altered morphologies. Original magnification: 600×. (**B**): In the DG of juvenile and adult mice, the numbers of IBA+ microglia cells were quantified after 20 × 0.1 Gy versus 1 × 2 Gy (72 h post-IR), in comparison to age-matched, sham-irradiated controls. Error bars represent SEM, n ≥ 3; * denotes statistically significant difference compared to sham-irradiated control: * *p* < 0.05; ** *p* < 0.01.

**Figure 4 cancers-14-05477-f004:**
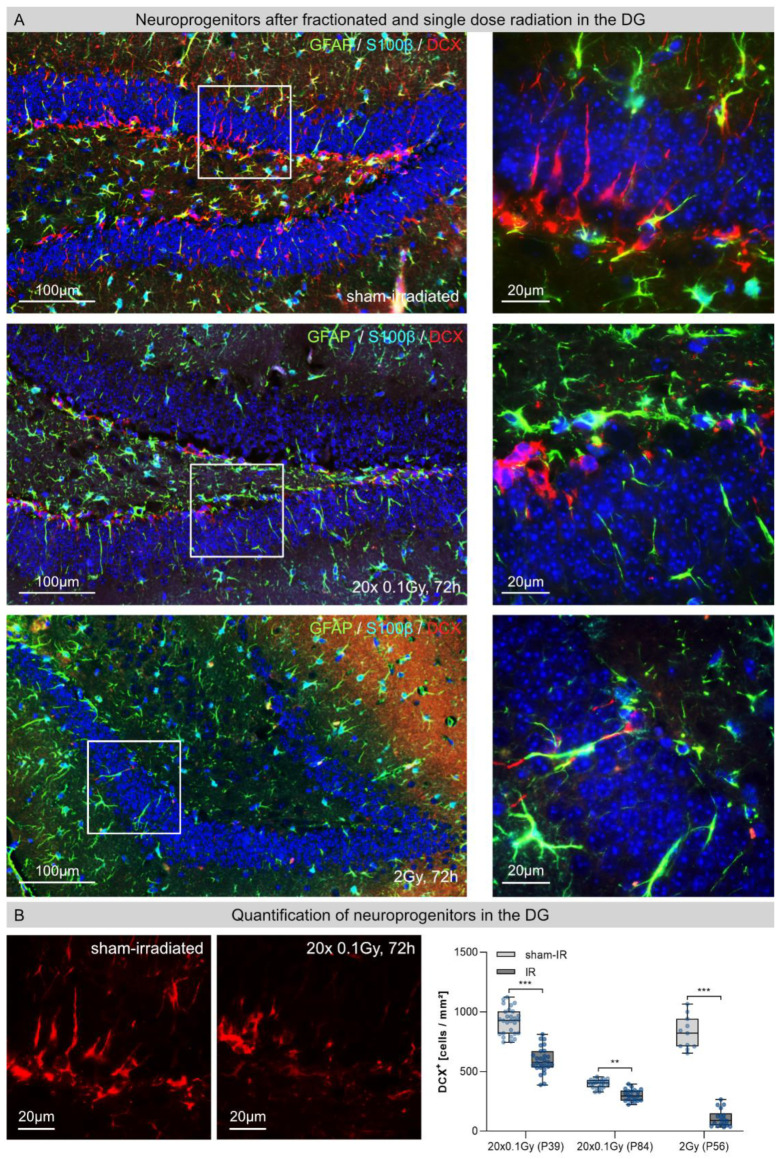
Neuroprogenitors in the hippocampal dentate gyrus (DG) after fractionated low-dose and single-dose radiation. (**A**): Immunofluorescence micrographs of GFAP+ (green)/S100β+ (light-blue) astrocytes and DCX+ (red) neuroprogenitors in the DG of sham-irradiated (upper panel), fractionated low-dose irradiated (20 × 0.1 Gy, 72 h post-IR; middle panel) and single-dose irradiated (1 × 2 Gy, 72 h post-IR; lower panel) brains. Framed regions are shown at higher magnifications. Insets: following FLDR, neuroprogenitors showed dramatically impaired dendritic arborization; following SDR, neuroprogenitor populations in the SGZ were drastically reduced. Original magnification: 600x. (**B**): In the DG of juvenile and adult mice, the numbers of DCX+ neuroprogenitors were quantified after 20 × 0.1 Gy versus 1 × 2 Gy (72 h post-IR), in comparison to age-matched, sham-irradiated controls. Error bars represent SEM, n ≥ 3; * denotes statistically significant difference compared to sham-irradiated control: ** *p* < 0.01, *** *p* < 0.001.

## Data Availability

All data generated and analyzed during this study are included in this published article.

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
