# Peer review of "Region-Specific Effects of Fractionated Low-Dose Versus Single-Dose Radiation on Hippocampal Neurogenesis and Neuroinflammation"

_cancers, 2022, doi:10.3390/cancers14225477_

Round 1

Reviewer 1 Report

In this manuscript, the authors reported similar results as they have published previously (below). No additional novel findings were provided.  

Schmal Z, Hammer B, Muller A, Rube CE. Fractionated Low-Dose Radiation Induces Long-Lasting Inflammatory Responses in the Hippocampal Stem Cell Niche. Int J Radiat Oncol Biol Phys 2021 300 12.

Schmal Z, Isermann A, Hladik D, von Toerne C, Tapio S, Rube CE. DNA damage accumulation during fractionated low-dose 301 radiation compromises hippocampal neurogenesis. Radiother Oncol 2019;137:45-54 302 13. 

Author Response

Reviewer #1

In this manuscript, the authors reported similar results as they have published previously (below). No additional novel findings were provided.

Schmal Z, Hammer B, Muller A, Rube CE. Fractionated Low-Dose Radiation Induces Long-Lasting Inflammatory Responses in the Hippocampal Stem Cell Niche. Int J Radiat Oncol Biol Phys 2021 300 12.

Schmal Z, Isermann A, Hladik D, von Toerne C, Tapio S, Rube CE. DNA damage accumulation during fractionated low-dose  radiation compromises hippocampal neurogenesis. Radiother Oncol 2019;137:45-54 302 13.

In contrast to our earlier work, we compared different hippocampal sub-regions, namely neurogenic dentate gyrus and non-neurogenic cornu ammonis and, and we used different fractionation schemes (20x 0.1Gy versus 1x 2Gy), to show the radiosensitivity as a function of neurogenesis. Here, we used SDR with 2Gy as this corresponds to the daily dose-fraction to target volumes of normo-fractionated radiotherapy. In addition, FLDR with 20x 0.1Gy was used to simulate the lower doses outside actual target volumes in the context of conformal radiotherapy. Basically, our results show that neurogenicity, i.e. proliferative capacity of neuronal progenitor cells, decisively determines radiation-induced inflammatory responses. This proliferative capacity is significantly higher in dentate gyrus than cornu ammonis, as well as in juvenile compared to adult brain. Although the cumulative dose is the same, almost all neuroprogenitors were eliminated in the dentate gyrus following SDR with 2Gy, while neuroprogenitors were significantly spared with FLDR (Fig. 4). These results underline the importance of fractionation in clinical radiotherapy, especially for proliferating neuroprogenitors. The increased radiation-induced elimination of neuroprogenitors in dentate gyrus is associated with increased inflammatory responses of microglia (Fig. 2) and astrocytes (Fig. 3). Overall, these current investigations expand previous radiobiological findings by showing that  hippocampal radiosensitivity is clearly attributed to the proliferative potential of neurogenesis.

Reviewer 2 Report

The authors investigate mechanisms underlying an important problem that faces clinicians who treat brain tumor patients, as well as these patients themselves and their friends and family. That is, that while effective at tumor control, brain-directed radiotherapy can cause significant cognitive dysfunction. The authors identify this problem clearly and present data to address the underlying cellular changes that occur in memory areas of the brain. Using immunofluorescence microscopy of irradiated brains of the C57Bl6/ncrl mouse strain, the authors catalogue cellular changes in the hippocampus following 2 different irradiation schedules in mice of different ages. While the data are interesting, and certainly would warrant further study, there are problems with the original design of the experiments and presentation of the data that severely limit the impact of the manuscript.

With respect to experimental design:

1) The rationale for the choice of the two different irradiation schedules appears to be that both represent a cumulative dose that is numerically the same. There is a well-known radiobiological relationship between dose rate and radiation effects that underlies the use of fractionation (related to the rate of repair of radiation-induced DNA damage). Two dose schedules of the same numerical cumulative dose but delivered at different dose rates would not be expected to produce similar results. This is especially true in terms of cytotoxic effects on cells like neural progenitor cells, as the survival effects of radiation dose fractionation are the most firmly established. A better design would have used equivalent biologically effective doses calculated using the linear-quadratic equation at a minimum. For example, using this equation and an alpha-beta ratio of 5 for neuroprogenitor cells, the single fraction dose would be expected to have 1.4 fold more cytotoxic effect, which may be close to what the data show in Figure 4. The contribution of the work therefore is limited in novelty, but rather recapitulates principles of radiation biology that are well-established.

2) The choice to evaluate the tissues at 72 hours after last dose of radiation makes comparisons between the dose schedules difficult, as the fractionated approach will necessarily involve more chronic changes, given that the course of radiation required 4 weeks to complete. This complicates comparisons, as the chronicity of the changes will confound any comparison.

3)Likewise, as cognitive effects of radiation are known to take months to manifest in humans, the utility of assays 72 hrs following a single dose of radiation are difficult to interpret, especially without some correlation to a more long-term cognitive endpoint.

4) The design to use mice of different ages at the start of fractionated treatment also has potential to be a confounding variable, as at the time of analysis, the ages of mice were all different across the fractionation regimens.

With respect to data presentation:

1) Any discussion of the effects on younger vs. older mice must be restricted to the fractionated irradiation schedule, as it seems that only one age of mice received single fraction dose.

2) It seems that any description of the data presented in Figure 4 is absent from the text: There is only the figure legend. Is this a formatting or editing error, either in the journal’s system for reviewers, or by the authors themselves? The effects of radiation on neuro progenitors is arguably the most interesting and pertinent data in the manuscript, and should be described much more thoroughly. 

Author Response

Reviewer #2

The authors investigate mechanisms underlying an important problem that faces clinicians who treat brain tumor patients, as well as these patients themselves and their friends and family. That is, that while effective at tumor control, brain-directed radiotherapy can cause significant cognitive dysfunction. The authors identify this problem clearly and present data to address the underlying cellular changes that occur in memory areas of the brain. Using immunofluorescence microscopy of irradiated brains of the C57Bl6/ncrl mouse strain, the authors catalogue cellular changes in the hippocampus following 2 different irradiation schedules in mice of different ages. While the data are interesting, and certainly would warrant further study, there are problems with the original design of the experiments and presentation of the data that severely limit the impact of the manuscript.

With respect to experimental design:

1) The rationale for the choice of the two different irradiation schedules appears to be that both represent a cumulative dose that is numerically the same. There is a well-known radiobiological relationship between dose rate and radiation effects that underlies the use of fractionation (related to the rate of repair of radiation-induced DNA damage). Two dose schedules of the same numerical cumulative dose but delivered at different dose rates would not be expected to produce similar results. This is especially true in terms of cytotoxic effects on cells like neural progenitor cells, as the survival effects of radiation dose fractionation are the most firmly established. A better design would have used equivalent biologically effective doses calculated using the linear-quadratic equation at a minimum. For example, using this equation and an alpha-beta ratio of 5 for neuroprogenitor cells, the single fraction dose would be expected to have 1.4 fold more cytotoxic effect, which may be close to what the data show in Figure 4. The contribution of the work therefore is limited in novelty, but rather recapitulates principles of radiation biology that are well-established.

We fully agree with the reviewer that using different fractionation schemes our findings on hippocampal neurogenesis reflect well-established fractionation effects. While 2Gy single-dose corresponds to the daily dose-fraction to target volumes of normo-fractionated radiotherapy, FLDR (20x 0.1Gy) may simulate the lower doses outside actual target volumes in the context of conformal radiotherapy.  Our results impressively show the well-established fractionation effects and emphasize the importance of lowering the dose-fraction for the vulnerable dentate gyrus in the context of clinical radiation planning.

Revised manuscript: Page 9, line 270: “While SDR with 2Gy corresponds to the daily dose-fraction to target volumes, FLDR (20x 0.1Gy) may simulate the repetitive lower doses outside actual target volumes in the context of normo-fractionated conformal radiotherapy… Collectively, these results impressively show the well-established fractionation effects and emphasize the importance of lowering the dose-fraction for the vulnerable dentate gyrus in the context of clinical radiation planning.”

2) The choice to evaluate the tissues at 72 hours after last dose of radiation makes comparisons between the dose schedules difficult, as the fractionated approach will necessarily involve more chronic changes, given that the course of radiation required 4 weeks to complete. This complicates comparisons, as the chronicity of the changes will confound any comparison.

In our earlier studies, we have already been able to show that chronic neuroinflammation persists for up to 6 months after FLDR. Of course, we agree with the expert that significantly more chronic effects play a role in the radiation response of FLDR. In future studies it would certainly be useful to investigate the chronic effects of SDR with 2Gy.

Revised manuscript: Page 9, line 273: “In earlier studies, we have already been able to show that chronic neuroinflammation persists for up to 6 months even following FLDR (11).”

3) Likewise, as cognitive effects of radiation are known to take months to manifest in humans, the utility of assays 72 hrs following a single dose of radiation are difficult to interpret, especially without some correlation to a more long-term cognitive endpoint.

Page 10 line 293: “In order to measure functional long-term damage to neurocognitive function, behavioral studies over longer periods of time should be carried out in future experiments.”

4) The design to use mice of different ages at the start of fractionated treatment also has potential to be a confounding variable, as at the time of analysis, the ages of mice were all different across the fractionation regimens.

Young and adult mice were deliberately examined in these studies, since in clinical radiotherapy immature brains of children are significantly more sensitive to radiation than mature brains of adults. Here, we can show one important reason for this increased radiosensitivity, namely the clearly stronger neurogenesis of the developing brain.

With respect to data presentation:

1) Any discussion of the effects on younger vs. older mice must be restricted to the fractionated irradiation schedule, as it seems that only one age of mice received single fraction dose.

The discussion regarding radiation-induced effects in juvenile versus adult mice has been limited to FLDR.

2) It seems that any description of the data presented in Figure 4 is absent from the text: There is only the figure legend. Is this a formatting or editing error, either in the journal’s system for reviewers, or by the authors themselves? The effects of radiation on neuro progenitors is arguably the most interesting and pertinent data in the manuscript, and should be described much more thoroughly.

We have to apologize for this mistake. Obviously, this crucial section fell out during formatting.

Page 6, line 199-216: “3.4. Neuroprogenitors in the dentate gyrus (DG) after fractionated low-dose and single-dose radiation: In sham-irradiated hippocampi numerous doublecortin-expressing (DCX+) neuroprogenitors in the SGZ of DG showed intact dendritic arborizations with processes extending through granular cell layers (Fig. 4A, upper row). In Fig. 4A representative micrographs of fractionated low-dose (20x 0.1Gy, 72h post-IR) and single-dose (2Gy, 72h post-IR) irradiated hippocampi are depicted to demonstrate the directly visible neurotoxic effects (Fig. 4A, lower rows). Following FLDR the hippocampal DG exhibited neuroprogenitors with dramatically impaired dendritic arborization (Fig. 4A second row), indicating the critical impact of even low doses on neuronal differentiation and maturation. Furthermore, the number of DCX+ neuroprogenitors was significantly reduced (juvenile P39: sham-IR: 915 ±52 cells/mm²; 72h post-IR: 593 ±23 cells/mm², ≈35% reduction; adult P84: sham-IR: 398 ±25 cells/mm²; 72h post-IR: 305 ±24 cells/mm², ≈25% reduction). Following SDR the population of neural progenitors in hippocampal DG was even more drastically reduced, so that nearly all cell bodies of DCX+ neuroprogenitors were completely abrogated (P56: sham-IR: 779 ±91, 72h post-IR: 109 ±29 cells/mm², ≈85% reduction) (Fig. 4A & B). Collectively, these findings demonstrate the fatal effects of ionizing radiation to hippocampal DG, leading to drastically reduced neurogenesis, neuronal differentiation and maturation.”

Reviewer 3 Report

This is an interesting study regarding a clinically relevant topic. I have the following comments:

1) Why was this particular radiation dose and fractionation chosen? Were there prior dose finding studies?

2) Mice were sacrificed 72 hours post-RT. Were further timepoints considered? If these are acute effects, would recovery be expected? Would repopulation of neuroprogenitors be possible? If these effects are so marked, and also permanent at a dose of 2Gy, there is little hope for hippocampal sparing radiotherapy, as it is nearly impossible to deliver therapeutic radiotherapy doses while limiting an organ at risk completely enclosed in target volume to such a low dose. If the authors anticipate this degree of loss of progenitors to result in a subclinical phenotype, as one might expect with such a low dose of radiotherapy, how do they explain the mismatch between degree of progenitor loss, and subclinical phenotype?

3) Please provide N for every bar graph and figure. Since there are so few mice/replicates, it would be helpful to simply plot every datapoint with superimposed box/whiskers, easy to do in Graphpad or other statistical plotting software. 

4) The N is very small for each claim made. The authors should explain why a small N was used, and why they feel the results should be considered robust when considering ~3 mice per condition. 

Author Response

This is an interesting study regarding a clinically relevant topic. I have the following comments:

1) Why was this particular radiation dose and fractionation chosen? Were there prior dose finding studies?

In this current work, the specific radiation sensitivity of different hippocampal sub-regions was investigated in the brain of juvenile and adult mice. In contrast to our earlier work, we compared the non-neurogenic cornu ammonis and the neurogenic dentate gyrus, and we used different fractionation schemes (20x 0.1Gy versus 1x 2Gy), to show the radiosensitivity as a function of neurogenesis. Here, SDR with 2Gy was used, corresponding to the single fraction of normo-fractionated radiotherapy to the target volume. In addition, FLDR with 20x 0.1Gy was used to simulate the lower doses outside the actual target volume in the context of conformal radiotherapy. Basically, these results show that the neurogenicity, i.e. the proliferative capacity of the neuronal progenitor cells, decisively determines the sensitivity to radiation. This proliferative capacity is significantly higher in the gyrus dentate than in the cornu ammonis, as well as in juvenile compared to adult brain. Although the cumulative dose is the same, almost all neuroprogenitors were eliminated in the dentate gyrus with SDR of 2Gy, while the neuroprogenitors are significantly spared with FLDR (Fig. 4). These results underline the importance of fractionation in clinical radiotherapy, especially for proliferating neuroprogenitors. The increased radiation-induced elimination of neuroprogenitors in the dentate gyrus is associated with increased inflammatory responses of microglia (Fig. 2) and astrocytes (Fig. 3). Overall, these current investigations expand the radiobiological findings regarding the specific radiosensitivity of the hippocampus region, whereas this radiation sensitivity is clearly attributed to the proliferative potential of neurogenesis in the dentate gyrus (Fig. 4).

2) Mice were sacrificed 72 hours post-RT. Were further timepoints considered?

In our earlier studies, we have already been able to show that chronic neuroinflammation persists for up to 6 months after fractionated LDR.

Schmal Z, Hammer B, Muller A, Rube CE. Fractionated Low-Dose Radiation Induces Long-Lasting Inflammatory Responses in the Hippocampal Stem Cell Niche. Int J Radiat Oncol Biol Phys 2021 300 12.

Page 9 line 273: “In earlier studies, we have already been able to show that chronic neuroinflammation persists for up to 6 months after FLDR (11).”

If these are acute effects, would recovery be expected? Would repopulation of neuroprogenitors be possible? If these effects are so marked, and also permanent at a dose of 2Gy, there is little hope for hippocampal sparing radiotherapy, as it is nearly impossible to deliver therapeutic radiotherapy doses while limiting an organ at risk completely enclosed in target volume to such a low dose.

According to the clinical observations, we expect a recovery of hippocampal neurogenesis for both, FLDR and SDR with cumulative 2Gy. In previous work, we were able to show that slowly proliferating SOX2+ stem/progenitor cells survive FLDR and may replace damaged or eliminated DCX+ neuroprogenitor cells through premature differentiation. However, our results underline the extreme radiosensitivity of neurogenesis; accordingly, the cumulative dose and single dose-fraction should be kept as low as possible.

Page 10, line 289: “In previous work, we were able to show that slowly proliferating SOX2+ stem/progenitor cells survive FLDR and may replace damaged or eliminated DCX+ neuroprogenitor cells through premature differentiation. Therefore, we expect at least partial recovery of hippocampal neurogenesis for both, FLDR and SDR with cumulative 2Gy.”

 If the authors anticipate this degree of loss of progenitors to result in a subclinical phenotype, as one might expect with such a low dose of radiotherapy, how do they explain the mismatch between degree of progenitor loss, and subclinical phenotype?

We assume that the healthy brain has excellent capacities to regenerate neurogenesis due to surviving SOX2+ stem cells. Only when these early stem/progenitor cells are eliminated by IR exposure in the long term, the ability to regenerate is significantly reduced.  

3) Please provide N for every bar graph and figure. Since there are so few mice/replicates, it would be helpful to simply plot every datapoint with superimposed box/whiskers, easy to do in Graphpad or other statistical plotting software. 

To generate consistent results 3 mice with their right and left hippocampi, each with 3 technical replicates (3 different tissue sections spaced 80 µm apart) were analyzed at each assessment time-point and compared to sham-irradiated age-matched controls (n= 3). Accordingly, each data point usually results from 18 different measurements. (Graphs now show the datapoints with superimposed box/whiskers.)

Page 3 line 107: “…right and left hippocampi of 3 different adult and juvenile mice (n= 3), each with 3 technical replicates (3 tissue sections spaced in 80 µm distance) were analyzed at each assessment time-point and compared to sham-irradiated age-matched controls.”

4) The N is very small for each claim made. The authors should explain why a small N was used, and why they feel the results should be considered robust when considering ~3 mice per condition. 

In our earlier work, 3 animals each were examined at different times during (after 5x, 10x, 15x, 20x 0.1Gy) and after fractionated LDR (1m, 3m, 6m post-IR)(in total: 21 juvenile and 21 adult mice compared to age-matched controls). This extensive data on the time course of the decrease in DCX+ and SOX2+ stem cell/progenitor cells after fractionated LDR shows very consistently the radiation effects on neurogenesis. Moreover, we analyzed DNA repair deficient rodent strains, such as Ataxia telangiectasia and SCID mice, which showed pronounced effects on neurogenesis even at very low doses. As a result, we were able to gain robust data and extensive experience regarding the effects of FLDR on neurogenesis and can therefore better classify new results, even with fewer data.

Schmal Z, Isermann A, Hladik D, von Toerne C, Tapio S, Rube CE. DNA damage accumulation during fractionated low-dose radiation compromises hippocampal neurogenesis. Radiother Oncol 2019;137:45-54 302 13.

Reviewer 4 Report

Well written manuscript on  influence of radiation dose and fractionation  on hippocampal neurogenesis in young and adult mice. This is not new and has been reported earlier by e.g. Redmond KJ et al J Neurooncol 2011 Sep;104(2):579-87 and Neuro Oncol 2013 Mar;15(3):360-9 or  Monje M et al Annal Neurolog 2007 not only in mice but also on human autopsies and in clinic. Confirmation of knowledge by a second group is always wanted, however, I would like to see those data from other groups being cited and discussed. Currently, they cite only their own previous work. Furthermore, there is a vast literature and data on clinical experience of hippocampal sparing  including several randomized trials. The authors cite only  a paper from Grosu et al on a planned randomized trial. This should be adapted. 

Next, I think that the method used to irradiate the whole mouse (whole body irradiation) is no longer really the best method in investigating radiation to the brain. Newer methods allow only radiotherapy of the brain inclusive detailed dose planning. This is also possible with a linear accelerator and will have influence on the dose distribution (smaller field dosimetry). 

Has this work  been performed by 2 persons only, are there not more  to be acknowledged in the author list. 

Detail: Page 2, line 54: the word "been" is missing, should be: ..not yet been analyzed in detail.

Author Response

Well written manuscript on influence of radiation dose and fractionation on hippocampal neurogenesis in young and adult mice. This is not new and has been reported earlier by e.g. Redmond KJ et al J Neurooncol 2011 Sep;104(2):579-87 and Neuro Oncol 2013 Mar;15(3):360-9 or  Monje M et al Annal Neurolog 2007 not only in mice but also on human autopsies and in clinic. Confirmation of knowledge by a second group is always wanted, however, I would like to see those data from other groups being cited and discussed. Currently, they cite only their own previous work. Furthermore, there is a vast literature and data on clinical experience of hippocampal sparing  including several randomized trials. The authors cite only a paper from Grosu et al on a planned randomized trial. This should be adapted.

We thank the reviewers for the interesting literature references and will discuss them in the discussion section of the revised manuscript. In addition, we will cite other clinical trials on randomized hippocampus-sparing whole-brain radiotherapy.

Page 10, line 296: “In clinical oncology prospective studies analyzed the relationship between RT dose to neural progenitor cell niches and neurocognitive dysfunction in pediatric patients before and 6-, 15-, and 27-months following completion of cranial RT (15). These clinical data demonstrated significant associations between increasing RT dose to the hippocampus and decline in neurocognitive skills following cranial irradiation.”

Page 10, line 308: “Currently, prospective clinical trials analyze the feasibility of these techniques in cancer patients with different cranial RT indications and document the oncological and neurocognitive outcome following hippocampus-sparing radiotherapy (19-23). However, many studies are still ongoing, so that a conclusive assessment is not yet possible.”

Next, I think that the method used to irradiate the whole mouse (whole body irradiation) is no longer really the best method in investigating radiation to the brain. Newer methods allow only radiotherapy of the brain inclusive detailed dose planning. This is also possible with a linear accelerator and will have influence on the dose distribution (smaller field dosimetry). 

We agree with the reviewer that, in principle, localized irradiation of the brain would be preferable. However, since FLDR is used almost daily for 4 weeks (except for the weekends), localized brain irradiation with the necessary anesthesia is very traumatizing, especially for the young mice, and leads to very high animal losses.

Has this work been performed by 2 persons only, are there not more to be acknowledged in the author list.

In fact, these investigations were carried out solely by the two authors. Technical support for animal irradiation is mentioned in the acknowledgment.

Page 10, line 330: “Acknowledgments: The authors acknowledge the technical assistance of Gargi Tewary for performing animal experiments, and of our technical staff at the linear accelerator for animal irradiation.”

Detail: Page 2, line 54: the word "been" is missing, should be: ..not yet been analyzed in detail.

The missing word has been added.